# A Case of Ertapenem Neurotoxicity Resulting in Vocal Tremor and Altered Mentation in a Dialysis Dependent Liver Transplant Patient

**DOI:** 10.3390/antibiotics8010001

**Published:** 2018-12-22

**Authors:** Ramy M. Hanna, Shih-Fan Sun, Pryce Gaynor

**Affiliations:** 1Division of Nephrology, Department of Medicine, David Geffen School of Medicine, University of California Los Angeles, 700 Tiverton Ave., Los Angeles, CA 90095, USA; 2Department of Medicine, David Geffen School of Medicine, University of California Los Angeles, 700 Tiverton Ave., Los Angeles, CA 90095, USA; Shih-Fansun@mednet.ucla.edu; 3Division of Infectious Diseases, Department of Medicine, David Geffen School of Medicine, University of California Los Angeles, 700 Tiverton Ave., Los Angeles, CA 90095, USA; Pgaynor@mednet.ucla.edu

**Keywords:** carbapenem, encephalopathy, vocal tremor, neurotoxicity

## Abstract

Carbapenem agents are advanced derivatives of cephalosporins that are active against bacteria that produce extended spectrum beta lactamases (ESBL). These antibiotics are resistant to enzymatic cleavage, and have good central nervous system penetration. Given this fact, it is not surprising that these drugs have been reported to cause neurological side effects like seizures and encephalopathy. We report a case of a patient on hemodialytic support who had a notable change in mentation and vocal tremor. This was at first attributed to calcineurin toxicity, but after the finding of a normal tacrolimus level, ertapenem neurotoxicity was suspected. After discontinuation of the offending agent, the patient’s vocal tremor, cognition, and neurological function returned to baseline levels.

## 1. Introduction 

Neurotoxicity is a potential side effect of many antibiotics that are capable of penetrating the blood brain barrier [1,2,3,4,5,6,7,8]. Carbapenems, in particular, can achieve therapeutic levels in the Central Nervous System (CNS), and as such have been associated with neurotoxicity [3,4,5,6,7,8]. The agents in this class include imipenem, ertapenem, doripenem, and meropenem [1,2], and are reserved for resistant infections [9]. Human studies have documented an epileptogenic potential with imipenem greater than the other carbapenems [10]. While the rate of seizures with non-imipenem carbapenems is reported at <1% [10,11], other neurologic toxicities such as encephalopathy have also been reported [3,4,5,6,7,8,10,11,12,13]. Adverse neurological events have also been reported in patients without renal insufficiency [3]. Murine studies have shown electroencephalogram changes without seizures in rats treated with imipenem [12]. This observation provides a mechanism explaining how carbapenems can cause encephalopathy [12]. We report a case of ertapenem neurotoxicity in an end stage renal disease patient on maintenance hemodialysis who was also an orthotopic liver transplant recipient. 

## 2. Case Report

We report a 74-year-old male with end stage liver disease who received an orthotopic liver transplant in March 2018 with improvement of his hepatic encephalopathy and synthetic function. He suffered from acute kidney injury shortly before the transplant due to suspected hepatorenal syndrome, and became dialysis dependent prior to his transplantation. He had been stably maintained on an outpatient regimen of dialysis every Monday, Wednesday, and Friday. He was listed for a renal transplant and was receiving hemodialysis through a tunneled central venous catheter rather than through arteriovenous access. 

The patient presented in October 2018 with weakness and chills and was found to be bacteremic with ESBL *Klebesiella oxytoca.* He was treated in August 2018 for a polymicrobial bacteremia, due to a catheter related infection. The blood cultures in August grew *Citrobacter*, *Stenotrophomonas* ESBL, and ESBL *Klebesiella oxytoca*. The ertapenem was dosed for hemodialysis and was to be administered with dialysis rather than daily. 

One week into therapy, the patient noted a tremor in his voice, weakness, confusion, forgetfulness, and his wife noted a change in his personality. Upon interview with his nephrologist, it was clear this once lucid patient had a profound change in his mentation and personality. Suspecting an interaction between the carbapenem and the prescribed calcineurin inhibitor (tacrolimus), he was sent back to Ronald Reagan Medical Center, UCLA for hospitalization workup and emergent measurement of tacrolimus levels. It was verified that the hepatic cytochrome 450 enzymatic system should not be affected by ertapenem [14]. The tacrolimus level was 5.3 ng/mL, which is within normal limits and at goal for the patient’s liver transplant. 

After the diagnosis of ertapenem neurotoxicity was suspected on clinical grounds, the drug was discontinued with ongoing hemodialysis on the regular Monday, Wednesday, Friday schedule. The patient returned to his baseline mental status rapidly after drug discontinuation. His blood cultures remained clear and he was able to continue use of his tunneled central venous dialysis catheter while awaiting renal transplantation. The patient is now amenable to placement of a permanent arteriovenous access since renal transplantation maybe delayed further due to catheter related infections and frailty.

## 3. Discussion

We report a case of a patient who developed vocal tremor, a change in mentation and level of attention, and a personality change in response to ertapenem neurotoxicity. This occurred despite the patient getting a reduced dose of ertapenem at 0.5 grams intravenously every other day due to the advanced level of renal dysfunction (ESRD patient) [15]. This case demonstrates that the potent CNS penetration of carbapenems can be expected to produce some unusual neurological manifestations besides seizures [3,4,5,6,7,8,9,10,11,12,13]. An accounting of the evidence using the Naranjo scale yielded a score of 7 which indicates that ertapenem administration was related to the observed neurotoxicity as a probable adverse drug reaction [16]. 

The differential diagnosis for drug-induced encephalopathy in liver transplant patients should include hepatic encephalopathy due to graft failure and calcineurin neurotoxicity in cases with extremely high serum trough levels of calcineurin inhibitors. This case illustrates an interesting presentation of ertapenem neurotoxicity in an ESRD patient.

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
