# Peer review of "A Case of Ertapenem Neurotoxicity Resulting in Vocal Tremor and Altered Mentation in a Dialysis Dependent Liver Transplant Patient"

_antibiotics, 2018, doi:10.3390/antibiotics8010001_

Round 1
Reviewer 1 Report
How can an idiosyncratic interaction between the two treatments be excluded?
line 68: ..potent CNS penetration of can be expected....a word is missing after of
Author Response
Reply to reviews for “antibiotics” manuscript
Thank you to the reviewers for their time and expertise. We have eagerly reviewed these suggestions and happily incorporated them into our manuscript.
Reply to reviewer 1:
line 68: ..potent CNS penetration of can be expected....a word is missing after of
Reply-Reviewed and wording changed. Thank you.
Reviewer 2 Report
Thank you for taking time to submit a manuscript describing your clinical findings. I strongly feel that case reports / series can provide valuable information to providers, esp for rare side effects. My recommendations include:
INTRO:
Line 25: The first subject you describe are carbapenems. That is not the subject. The subject is ertapenem or Neurotoxicity.
Line 26: If your intent is to describe carbepenems, doripenem is omitted.
Line 26 and 27: Brief discussion on use for resistant gram negative (line 26) and ESBL (line 26 / 27), although this a true statement - there are two things to consider: 1) The indications are much broader than resistant gram negative infections, especially ertapenem which is the focus of your paper (although I agree they should be reserved for such infections); 2) the focus of your paper is not resistant infections - it is side effects of ertapenem. A intro discussion the CNS side effects focusing on: J Antimicrob Chemother 2000;45:5-7; Pharmacotherapy. 2011 Apr;31(4):408-23; Antimicrobial Agents and Chemotherapy 1993;37:199-202; and especially Journal of Antibiotics 1995:48:408-416.
Line 32: Utilization of references 3-8. This is somewhat misleading. Although the statement is correct, the sentence needs to highlight that reports for your main subject have also been published. As currently written, it infers (at least to me) that ertapenem is void of CNS activity / side effects. I also think the definition of CNS needs to be changed. Somewhat of a switch between terms of CNS (line 29) and epileptogenic potential (line 31). Seizures is an example of a CNS reaction, but there are more. Also, other references for ertapenem neurotoxicity need to be included: Intern Med J 2014; 44(8):817-819; Am J Phys Med Rehabil. 2006; 85(3):267.
Line 44 and others: Klebesiella needs to be in italics and oxycota needs a) to be spelled oxytoca; b) in italics; c) O must be lower case.
Line 44: bacteria name in italics
Line 45: "same". How do you know it was the same and not a different colony type?
Line 66: how do you know this was the correct dosing of ertapenem? Did you check ertapenem levels via send out? Per the package insert - the recommended dosing is: "Adult patients with severe renal impairment (creatinine clearance 30 mL/min/1.73 m2 ) and end-stage renal disease (creatinine clearance 10 mL/min/1.73 m2 ) should receive 500 mg daily." Reference: https://www.accessdata.fda.gov/drugsatfda_docs/label/2012/021337s038lbl.pdf
Also - correct dosing cannot be determines unless calculating by first-order kinetics of the medication. For example, how did you incorporate the patients volume of distribution into the dose of the medication. Therefore, you cannot state you utilized the correct dose: 1) it is not correct via package insert; 2) you did not check levels; and 3) no mention of the patients size or expected Vd.
Line 69: sentence is incorrect. You stated besides the reported seizures with imipenem, but reference 3 is not for imipenem - it is for ertapenem. And other ertapenem publications (listed above in my peer review) where not mentioned.
Overall: Case reports publications should utilize the Naranjo scale: Naranjo CA, Busto U, Sellers EM, Sandor P, Ruiz I, Roberts EA, et al. A method for estimating the probability of adverse drug reaction. Clin Pharmacol Ther 1981;30:239–45.
Author Response
Reply to reviews for “antibiotics” manuscript
Thank you to the reviewers for their time and expertise. We have eagerly reviewed these suggestions and happily incorporated them into our manuscript.
Reply to reviewer 2:
Line 25: The first subject you describe are carbapenems. That is not the subject. The subject is ertapenem or Neurotoxicity.
Reply-This has been removed.
Line 26: If your intent is to describe carbepenems, doripenem is omitted.
Reply-Doripenem has been added.
Line 26 and 27: Brief discussion on use for resistant gram negative (line 26) and ESBL (line 26 / 27), although this a true statement - there are two things to consider: 1) The indications are much broader than resistant gram negative infections, especially ertapenem which is the focus of your paper (although I agree they should be reserved for such infections); 2) the focus of your paper is not resistant infections - it is side effects of ertapenem. A intro discussion the CNS side effects focusing on: J Antimicrob Chemother 2000;45:5-7; Pharmacotherapy. 2011 Apr;31(4):408-23; Antimicrobial Agents and Chemotherapy 1993;37:199-202; and especially Journal of Antibiotics 1995:48:408-416.
Reply-A short introduction to these CNS toxicity events will be included-with an eye towards the manuscript type and legth limitations, and these excellent papers will be included as references.
Line 32: Utilization of references 3-8. This is somewhat misleading. Although the statement is correct, the sentence needs to highlight that reports for your main subject have also been published. As currently written, it infers (at least to me) that ertapenem is void of CNS activity / side effects. I also think the definition of CNS needs to be changed. Somewhat of a switch between terms of CNS (line 29) and epileptogenic potential (line 31). Seizures is an example of a CNS reaction, but there are more. Also, other references for ertapenem neurotoxicity need to be included: Intern Med J 2014; 44(8):817-819; Am J Phys Med Rehabil. 2006; 85(3):267.
Reply-I believe inclusion of the suggested references and a limited discussion helped clarify this point. I added the above reference as well. I cannot find the last reference on pubmed, a more recent report of ertapenem toxicity (sutton et.al. was added in its stead).
Line 44 and others: Klebesiella needs to be in italics and oxycota needs a) to be spelled oxytoca; b) in italics; c) O must be lower case.
Reply-Done.
Line 44: bacteria name in italics
Reply-Done.
Line 45: "same". How do you know it was the same and not a different colony type?
Reply-Agree with this point, we have rephrased the from text.
Line 66: how do you know this was the correct dosing of ertapenem? Did you check ertapenem levels
via send out? Per the package insert - the recommended dosing is: "Adult patients with severe renal impairment (creatinine clearance £30 mL/min/1.73 m2 ) and end-stage renal disease (creatinine clearance £10 mL/min/1.73 m2 ) should receive 500 mg daily." Reference: https://www.accessdata.fda.gov/drugsatfda_docs/label/2012/021337s038lbl.pdf
Reply-Agreed, rephrased.
Also - correct dosing cannot be determines unless calculating by first-order kinetics of the medication. For example, how did you incorporate the patients volume of distribution into the dose of the medication. Therefore, you cannot state you utilized the correct dose: 1) it is not correct via package insert; 2) you did not check levels; and 3) no mention of the patients size or expected Vd.
Reply-Agreed, rephrased.
Line 69: sentence is incorrect. You stated besides the reported seizures with imipenem, but reference 3 is not for imipenem - it is for ertapenem. And other ertapenem publications (listed above in my peer review) where not mentioned.
Reply-Agreed rephrased.
Overall: Case reports publications should utilize the Naranjo scale: Naranjo CA, Busto U, Sellers EM, Sandor P, Ruiz I, Roberts EA, et al. A method for estimating the probability of adverse drug reaction. Clin Pharmacol Ther 1981;30:239–45.
Reply-Thank you for that excellent suggestion, on the naranajo scale we caluclated a score of 7 indicating a probable adverse event and cited above reference.
Round 2
Reviewer 2 Report
Only comment: Better. The following sentence is still mis-leading" "This case demonstrates that the potent CNS 71 penetration of carbapenems can be expected" This cases demonstrates for ertapenem. The data does not support a comment like this for the entire class of carbapenems. Recommend to accept.